# Life history optimisation drives latitudinal gradients and responses to global change in marine fishes

**Mariana Álvarez-Noriega**[1]*, **Craig R. White**[1], **Jan Kozłowski**[2], **Troy Day**[3,4], **Dustin J. Marshall**[1]

**1** Centre of Geometric Biology, School of Biological Sciences, Monash University, Melbourne, Victoria, Australia, **2** Institute of Environmental Sciences, Jagiellonian University, Kraków, Poland, **3** Department of Mathematics and Statistics, Queen's University, Kingston, Ontario, Canada, **4** Department of Biology, Queen's University, Kingston, Ontario, Canada

\* m.alvareznoriega@aims.gov.au

**Citation:** Álvarez-Noriega M, White CR, Kozłowski J, Day T, Marshall DJ (2023) Life history optimisation drives latitudinal gradients and responses to global change in marine fishes. PLoS Biol 21(5): e3002114. https://doi.org/10.1371/journal.pbio.3002114

**Data Availability Statement:** All relevant data are within the paper and its Supporting Information files.

## Abstract

Within many species, and particularly fish, fecundity does not scale with mass linearly; instead, it scales disproportionately. Disproportionate intraspecific size–reproduction relationships contradict most theories of biological growth and present challenges for the management of biological systems. Yet the drivers of reproductive scaling remain obscure and systematic predictors of how and why reproduction scaling varies are lacking. Here, we parameterise life history optimisation model to predict global patterns in the life histories of marine fishes. Our model predict latitudinal trends in life histories: Polar fish should reproduce at a later age and show steeper reproductive scaling than tropical fish. We tested and confirmed these predictions using a new, global dataset of marine fish life histories, demonstrating that the risks of mortality shape maturation and reproductive scaling. Our model also predicts that global warming will profoundly reshape fish life histories, favouring earlier reproduction, smaller body sizes, and lower mass-specific reproductive outputs, with worrying consequences for population persistence.

Reproductive allometry occurs when reproductive output increases disproportionately with body size. Reproductive allometry (as opposed to isometry) has recently been shown to be ubiquitous in marine fishes for batch fecundity [1], but it can also occur in other taxa ([2,3]; but see [4]), presenting challenges to our basic understanding of biological growth, how we manage disease vectors [3], and the way we use biological resources [5]. Most growth and life history models assumed that reproduction scales linearly with body size and failed to predict the existence of reproductive hyperallometry [1,2,6,7]. From a practical perspective, failing to account for reproductive hyperallometry risks underestimating the contribution of larger females to population replenishment [1,8] and results in massive, systematic errors in calculations of sustainable harvests [5].

One branch of theory–life history optimisation models [9,10] anticipated reproductive hyperallometry, but this prediction has largely been overlooked. In these models, the ideal

**Funding:** MAN was supported by the Centre for Geometric Biology, Monash University (https://cgb.org.au/). DJM was supported by a Future Fellowship (FT180100257) from the Australian Research Council. The funders had no role in study design, data collection and analysis, decision to publish, or preparation of the manuscript.

**Competing interests:** The authors have declared that no competing interests exist.

**Abbreviations:** GAM, general additive model; MPA, marine protected area; SST, sea surface water temperature.

resource allocation strategy is the one that maximises fitness for a given set of environmental conditions (e.g., for a given mortality regime). Like all models, life history optimisation models are a simplified representation of reality, and they use phenomenological descriptions of the mechanisms underlying the physiological constraints, leading to different energy intakes, usages, and allocation patterns. However, this simplification allows the models to focus on the determination of the best strategy of resource allocation for a given set of ecological conditions and physiological constraints [11]. While many studies investigating broad-scale biological patterns have focused on the effects of physiological constraints [12–14], far fewer have examined these patterns from an evolutionary optimisation perspective. Yet our capacity to predict how organisms will respond to human-mediated change depends critically on our knowledge of how organisms evolve and adapt to these conditions.

Life history models predict that reproductive scaling should be a product of mortality regimes and growth rates. Here, we conduct such a test for marine fishes—a group for which there are important practical reasons for understanding and predicting reproductive scaling [1,5,8]. Because fish life histories are known to vary along a latitudinal gradient [15,16], we gathered information on biogeographical variation in the key parameters that shape predictions of classic life history optimisation models for fish across latitudes: offspring size ($n = 52$, 47 species), rates of growth ($n = 52$, 47 species), and adult mortality ($n = 85$, 77 species). We then used these inputs to parameterise Day and Taylor's [9] life history evolution model to make predictions about how both reproductive schedules and reproductive scaling should covary with latitude. Finally, we tested these predictions by gathering data on age at 50% maturity ($A_{50}$, $n = 127$, 59 species) and fecundity–mass relationships ($n = 13,760$, 153 species) across the globe.

## Results and discussion

### Global patterns in mortality, growth, and offspring size

Natural mortality ($M$) (expressed as a size-independent mortality rate that is not a product of fishing) is highest at the tropics and decreases by 80% closer to the poles (from 0.85 $y^{-1}$ at 0˚ to 0.17 $y^{-1}$ at 60˚), which constitutes nearly a 2-fold increase in yearly survival probabilities (from 0.43 at 0˚ to 0.84 at 60˚, 95% CI 1.33- to 3.69-fold) ($R^2 = 0.16$) (S1A Fig). The effect of latitude on mass-specific growth rate ($k$) could not be differentiated from zero (the slope of $\ln(k)$ with latitude had an estimate of −0.01, 95% CI: −0.03 to 0.01; S1 Table), although the tendency was for $k$ to decrease with increasing latitude (from 4.36 $g^{1/4}$ $y^{-1}$ at 0˚ to 2.11 $g^{1/4}$ $y^{-1}$ at 60˚) ($R^2 = 0.13$) (S1B Fig). We did not detect an effect of latitude on the theoretical initial offspring weight ($w_0$) (similar to $t_0$ in the von Bertalanffy equation) (the slope of $\ln(w_0)$ with latitude had an estimate of 0.00, 95% CI: −0.5 to 0.2), with the median expectation for $w_0$ ranging between 12.9 and 14.4$g$ ($R^2 = 0.14$) (S1C Fig). Using these estimated relationships (with their attendant uncertainty), we calibrated the life history model to predict optimal reproductive schedules (see Methods).

### Model predictions: Patterns in reproductive schedules and scaling

Age at maturity was predicted to increase from the tropics to the poles (reproduce in the first year at 0˚ and 12.1 y at 60˚; Fig 1A). Reproductive scaling was predicted to triple across the same latitudes (2.8 at 0˚ to 8.7 at 60˚; Fig 2A). Consequently, reproductive scaling and age at maturity were predicted to positively covary with each other. Similar predictions arise under the assumption of size-dependent mortality (S2 Fig).

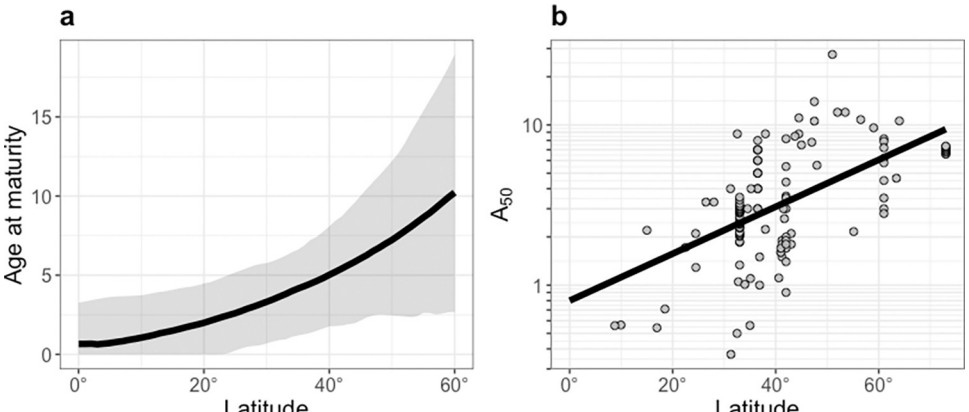

**Fig 1. Predicted and observed maturity schedules across latitudes.** (**a**) Optimal age at maturity predicted across latitudes. (**b**) Observed age at 50% maturity ($A_{50}$) (years) across latitudes. The solid line shows the statistical fit to the compiled data (Ln($A_{50}$) = $a$ + $b$ * latitude (absolute value), where $a$ is −0.28 (95% CI: −2.10 to 1.57), and $b$ is 0.03 (95% CI: −0.03 to −0.02); S1 Table). The grey points show the compiled data for age at 50% maturity (found in S1 Data). Note that the scale is arithmetic in (**a**), and logarithmic in (**b**).

## Testing the predictions: Global patterns in reproductive schedules and scaling

We found qualitative support for the predictions of our model when we compiled the available data from the literature. Age at maturity is difficult to measure, but a proxy for maturity, age at which 50% of individuals become mature ($A_{50}$), increased 7.7-fold across the same range of latitudes (0.76 y at 0° to 5.87 at 60°; $R^2$ = 0.84, 95% CI: 0.78 to 0.88) (Fig 1B).

When we examined the available batch fecundity data for 1,000s of individuals in over 100 species, we find support for the model predictions. Intraspecific reproductive scaling—the slope of the relationship between fecundity and mass (in log–log scale) for individuals of the

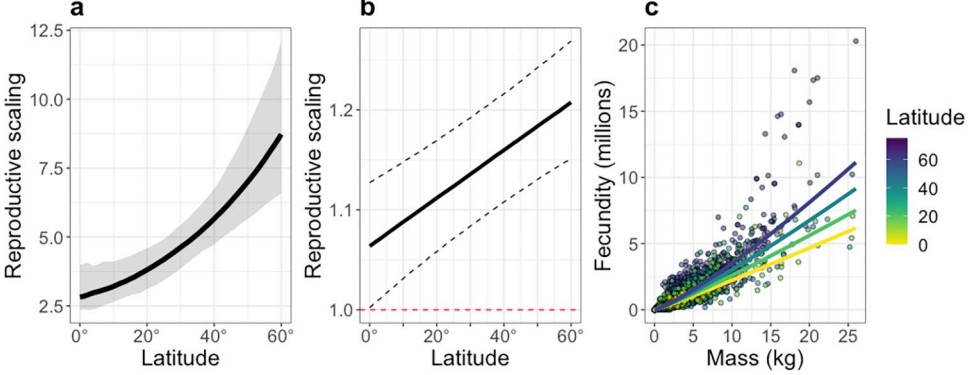

**Fig 2. Predicted and observed fecundity–mass relationships across latitudes.** (**a**) Reproductive scaling resulting from the predicted optimal age at maturity across latitudes. The black line shows the median of the predictions, and the grey ribbon shows the 95% credible intervals. (**b**) Observed mass scaling exponent of the number of eggs per female (RS) across latitudes (RS = $a$ + $b$ * latitude (absolute value), where $a$ is 1.00 (95% CI: 0.94 to 1.07) and $b$ is 0.004 (95% CI: 0.003 to 0.005); S1 Table). The black line shows the model estimates, and the dashed lines show the 95% credible intervals; the dotted red line shows an isometric scaling (i.e., when the reproductive scaling is 1). (**c**) The observed relationship of female mass and number of eggs ($F$) for different latitudes ($F$ = $c$ + $d$ * latitude + RS * mass, where $c$ is 5.39 (95% CI: 4.95 to 5.85), $d$ is −0.02 (95% CI: −0.03 to −0.02), and RS is calculated as for (**b**); S1 Table). The points show the data corrected for phylogeny, and the lines show the fitted model. The colours indicate absolute latitude from 0° (yellow) to 75° (purple). The underlying data for (**b**) and (**c**) can be found in S2 Data.

same species differing in size—is always greater than 1 and increases with latitude (13% increase from 0° to 60°; Fig 2B). For example, a 10-kg fish is predicted to produce approximately 2.2 million eggs at 0° latitude (0.5 to 9.2 million, 95% CI) and approximately 3.5 million at 60° latitude (0.9 to 14.4 million, 95% CI), a 20-kg fish is predicted to produce approximately 4.7 million eggs at 0° latitude (1.2 to 19.1 million, 95% CI) and approximately 8.1 million at 60° latitude (2.2 to 14.4 million, 95% CI). The number of eggs per female is higher for high-latitude species relative to low-latitude species across all female sizes except for the smallest females, where fecundities are essentially equivalent (Fig 2C). Because only a few species had data across a range of latitudes, different species were analysed at different latitudes, and, therefore, the dependence of within-species reproductive scaling on latitude must be treated as interspecific in Fig 2. For those species where mass and fecundity data were available across at least 20 degrees latitude, reproductive scaling also increased with latitude within species (S3 Fig). In contrast, when the relationship between mass and fecundity is estimated by taking a snapshot of the within-species relationship (e.g., only mass and fecundity at maturity instead of fecundity across all mature sizes within a population) and comparing this snapshot across species, such interspecific relationship is typically hypoallometric [6,17].

Although the direction of the relationship between reproductive scaling and latitude is successfully predicted by life history optimisation, the values of reproductive scaling estimated from the data are markedly lower than those predicted by the life history model. We suspect this discrepancy arises for 2 reasons: (i) how reproductive output is estimated; and (ii) how reproductive costs are modelled. First, our model makes predictions about the scaling of total reproductive output while we test these predictions based on the available data, which includes only 1 component of reproduction (batch fecundity). As well as having higher fecundity, larger females also tend to spawn more frequently and produce larger, heavier eggs than smaller females [18], so the scaling of total reproductive output (which incorporates the effects of spawning frequency, egg size, and batch fecundity) will be steeper than estimates based on batch fecundity alone. For example, *Sardiops sagax* shows a batch fecundity scaling of only approximately 1, but the scaling of total reproductive output in this species is approximately 3.5 because larger females spawn much more frequently than smaller females [18]. Hence, more comprehensive estimates of the scaling of total reproductive output are likely to be closer to those predicted by our model. Second, our model assumes all of the energy that goes into reproduction manifests as reproductive output, but in reality, a nontrivial proportion of reproductive allocation is likely to be expended as "overhead." While the overheads costs of reproduction are poorly resolved empirically, initial theoretical explorations show that including such costs would lower our predicted estimates of reproductive scaling and bring them closer to those observed in nature (S4 Fig). Empirically estimating how energy allocated into reproduction translates to reproductive output across different body sizes is necessary for better quantitative predictions of size–fecundity relationships.

While we acknowledge that our predictions differed quantitatively from the observed patterns, it is striking that a simple life history optimisation model [9], parameterised only with data on mortality, somatic growth, and offspring size across latitudes, can successfully predict qualitative global patterns in maturation and reproductive scaling for marine fish. Indeed, even when the effect of offspring size is neglected, the model still recovers the direction of change in age at maturity and reproductive scaling across latitudes. Broadly, and in relative terms, tropical fish suffer high mortality and maximise their fitness by diverting energy into reproduction earlier in life, reaching smaller sizes, having only shallow reproductive scaling exponents (though they are still greater than 1), and producing fewer eggs per unit mass. In contrast, polar fish experience lower mortality, and, therefore, their optimal strategy is to continue growing until later in life, reaching a larger size, and having steep reproductive scaling.

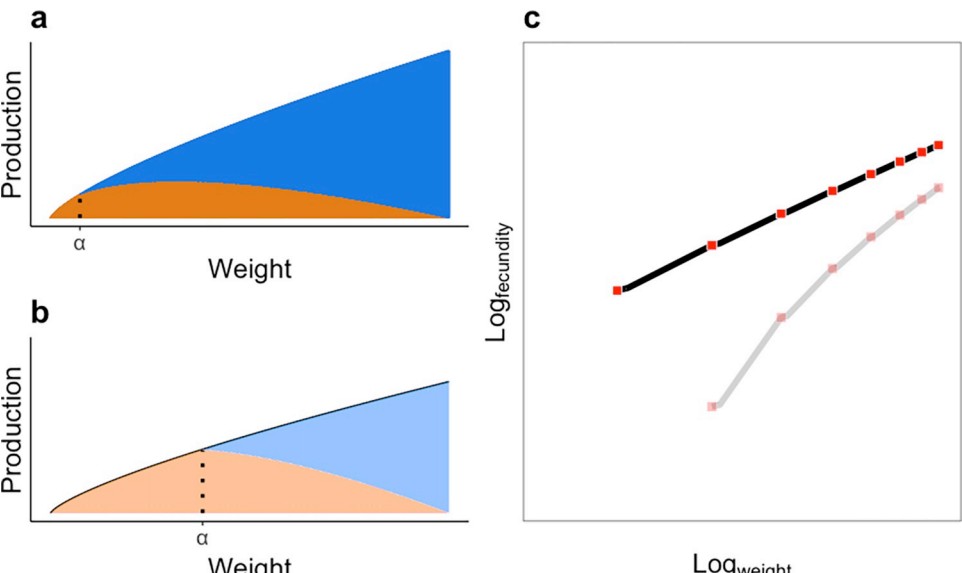

**Fig 3. Schematic demonstrating how reproductive hyperallometry increases with delayed maturity.** (**a** and **b**) Change in production and resource allocation as a function of size. Orange and blue areas show production allocated to growth and reproduction, respectively, and the opacity of the colours reflect early (solid) or late (translucid) maturity. The dotted vertical lines show the respective age at maturity ($\alpha$). Panel (**a**) shows predictions for early maturity (20° latitude), and panel (**b**) shows predictions for late maturity (40° latitude). For both latitudes, the scaling of the exponential decline in energy allocation for growth is the same, but the production curve is lower in panel (**b**) to reflect lower estimates of $k$ at high latitudes. (**c**) The predicted relationship between weight and fecundity (log–log scale) for fish maturing earlier (black line) and later (grey line). The red and pink squares show yearly-steps for early and late maturating fish, respectively. A steeper slope indicates higher reproductive scaling. These predictions were generated using Eqs 1–4 ($\alpha = 1.99$, $k = 3.41$, $w_0 = 13.87$ at 20°; $\alpha = 5.02$, $k = 2.68$ $w_0 = 13.35$ at 40° latitude).

In other words, in a fast–slow continuum, low-latitude fish tend to have fast life histories, whereas high-latitude fish tend towards the slow-paced end of the spectrum. These results contradict the expectation that highly seasonal environments should select for faster life history strategies, while in stable environments—such as the tropics—it would be beneficial to tend towards a slow-paced demography [19]. For example, low-latitude birds tend to show higher parental care and delayed maturation than temperate birds [20,21]. This discrepancy with our findings may be explained by patterns in mortality: in birds, mortality increases with latitude, as many die when temperatures drop in winter [22]. Our findings, in conjunction with others [11], imply that many of the life history patterns we observe in nature are driven, not only by mechanistic constraints, but also by the evolutionary optimisation of life histories. At the very least, life history optimisation approaches may be a useful tool for explaining patterns in life history based only on a few key parameters.

Hyperallometric scaling occurs because, as an increasing proportion of energy is diverted from growth to reproduction with time, the increments of female size decrease with age while the increments of reproductive output increase. Our model makes the novel prediction that fishes delaying reproduction because of low mortality produces a higher reproductive scaling. This pattern arises because the delay in the onset of reproduction yields larger sizes at maturity, and a shallower rate of increase in the size–production relationship whenever the scaling of production is <1 (Fig 3A and 3B). Hence, when a fish that has delayed reproduction finally starts allocating resources into reproduction, proportional changes in size between time-steps are small. If energy allocation from growth to reproduction changes with age rather than size, the amount of production allocated to reproduction increases more rapidly with size than for a

fish that matures early. For example, for a fish in the tropics, the total amount of production allocated into reproduction increases slowly across sizes (Fig 3A). In contrast, for a fish at high latitudes, while the postmaturity change in energy allocation with time is the same, the change with respect to size occurs much more abruptly (Fig 3B). Hence, the rate of increase in reproduction is much greater in a relative sense than the rate of growth compared to fish that mature early (Fig 3C). This means that reproductive scaling should be steeper in organisms that delay reproduction for longer—the compiled data support this prediction: higher-latitude species have later reproduction and higher reproductive scaling. Importantly, our model makes no predictions about how reproductive allocation should be divided in terms of offspring size and number though other models make such predictions [23]. Given that higher-latitude species tend to produce larger offspring, it would be interesting to see whether both fecundity and offspring size show steeper scaling at higher latitudes.

A steeper fecundity scaling in cold temperatures also occurs in taxa other than fish (e.g., flies [24], water fleas [25], and water snails [26]), but with exceptions (e.g., mosquitoes [27] and water striders [28]). A previous life history optimisation model from Arendt [29] argues that the size–temperature relationship, where organisms living in colder environments reach larger sizes than organisms in warm areas, can be explained by a steeper size–fecundity relationship in cold environments. If large organisms have a greater advantage in terms of fecundity, it may be optimal to delay maturation and reach a larger size. Here, we make the inverse argument: that delaying maturity is optimal when mortality is low, and the steeper reproductive scaling is the product of delaying reproduction. It is important to note that, although the changes in mortality and growth with latitude are certainly affected by a temperature gradient, other factors, such as light and predation [30], are likely influencing these patterns too. Experimental evolution studies seem ideally suited to teasing apart these 2 hypotheses: for example, multigenerational, orthogonal manipulations of mortality rate and temperature, and measuring life history changes (e.g., maturation schedules, size, and fecundity scaling) should provide a test for whether mortality drives life history evolution or temperature effects drive fecundity and size evolution. Interestingly, these data may already exist for some species but have not been analysed in this context [31].

In terms of fecundity, the average tropical fish produces fewer offspring than the average polar fish, and this disparity increases with female size such that larger, colder fish produce far more offspring for a given female mass than smaller, warmer fish. These results are also consistent with life history patterns of freshwater species [32,33] as well as other marine studies [34]. This is not to say that higher-latitude fish necessarily have higher fitness, as contrasting life history strategies may yield similar population growth in different environments [35]. The congruence between the model predictions and the compiled reproductive data suggests that reproductive schedules might well be a product of life history optimisation given local demographic or environmental constraints, particularly those affecting mortality. It is worth noting that in our model, we consider mortality as the driver of resource allocation, yet it is equally likely that different life histories will result in different levels of mortality-determining causality is difficult. A useful illustration of this point comes from a recent synthesis whereby mortality rates are the product of life history strategies [23]. Certainly, there is evidence that mortality can shape life history evolution [36] and that life history strategies affect mortality (e.g., in semelparous species). We suspect that causality is bidirectional—there is an interplay between mortality and life history, but reconciling these effects into the same modelling framework represents a formidable challenge.

Hyperallometric reproduction has important consequences for the replenishment of fish populations and how they are managed [37]. Fishing practices tend to disproportionally remove older, larger fish (i.e., "age-truncation"; [38,39]), shifting population size structure

towards smaller sizes. Therefore, reductions in spawning biomass through harvesting can have disproportionate consequences on reproductive output [40,41]. Our model predicts that accelerated mortality schedules associated with harvesting should lead to the evolution of lower mass-specific reproductive outputs—with concomitant losses in replenishment. Similarly, reproductive scaling is likely to strongly affect the spillover benefits of marine protected areas (MPAs) [8]. Both our model and data compilations show that high-latitude fisheries are likely to show steep reproductive scaling relative to the tropics. The degree to which reproductive hyperallometry matters for both fisheries models and MPAs is likely to systematically vary with latitude—while hyperallometry is ubiquitous, it is particularly pronounced near the poles. We would therefore predict that MPAs in polar regions, in particular, are likely to yield the greatest benefit for exploited populations.

If the latitudinal patterns observed here are mostly driven by temperature, global warming will reshape fish life histories and their demography. The general expectation is that fish will shrink in response to warming oceans [42–45] and that the productivity of most fisheries will decline with rising temperatures [46]. However, patterns in fish sizes associated to climate change are variable in nature [47]; tropical species have shrunk with increasing temperature, while temperate species show much more variable responses [48]. Assuming that temperature drives latitudinal changes in mortality, growth, and offspring size, our results suggest that regardless of how fish sizes change with increasing temperatures, mass-specific reproductive outputs are still likely to decline. We find that fish in warmer waters have lower mass-specific reproductive output than fish in cooler waters, and this discrepancy is particularly pronounced in larger fish. If future populations show the same temperature relationships as we observe here, a given standing spawner biomass of fish in the future will have much lower reproductive outputs than that same biomass today. For example, if $CO_2$ emissions remain high (i.e., an increase of 2.58˚C in sea surface temperature; [49]), our model predicts that a 25-kg fish at 60˚ latitude will produce about 300,000 fewer eggs by the end of the century, a difference of 5% of its mass-specific fecundity (Figs 4, S5 and S6). Such changes would have worrying consequences for replenishment and, thus, the productivity of global fisheries and may impact the food web in ways that are difficult to anticipate [50]. However, these predictions critically depend on all the inputs (offspring size, growth, and mortality) changing with temperature in the same ways as seen in a latitude gradient—an untested, yet critical assumption. Certainly, temperature is not the only factor that affects fecundity and changes across latitudes, and the effects of climate change on fish reproduction will depend on the joint effects of climate change on all drivers.

Predictions of marine fish reproductive schedules using life history optimisation prove similar to patterns observed in the compiled data. A valuable extension of the approach taken here would be to resolve the other parameters that likely change with latitude. For example, processes affecting resource acquisition and transformation to biomass and how they change with temperature were not considered here. Particularly, we expect the scaling of production to be temperature dependent, hence affecting the optimal life history strategy at each latitude. As an essential next step, we need a better understanding of how production scales with size, and how this relationship is affected by temperature.

## Materials and methods

We use a life history model proposed by Day and Taylor [9] to investigate the reproductive scaling arising from optimising energy allocation between growth and reproduction across ontogeny. In the model, growth follows different trajectories before and after maturity, and the change in energy allocation is determined by a continuous function. Investment into reproduction is initially zero and increases through time after maturity as determined by 2

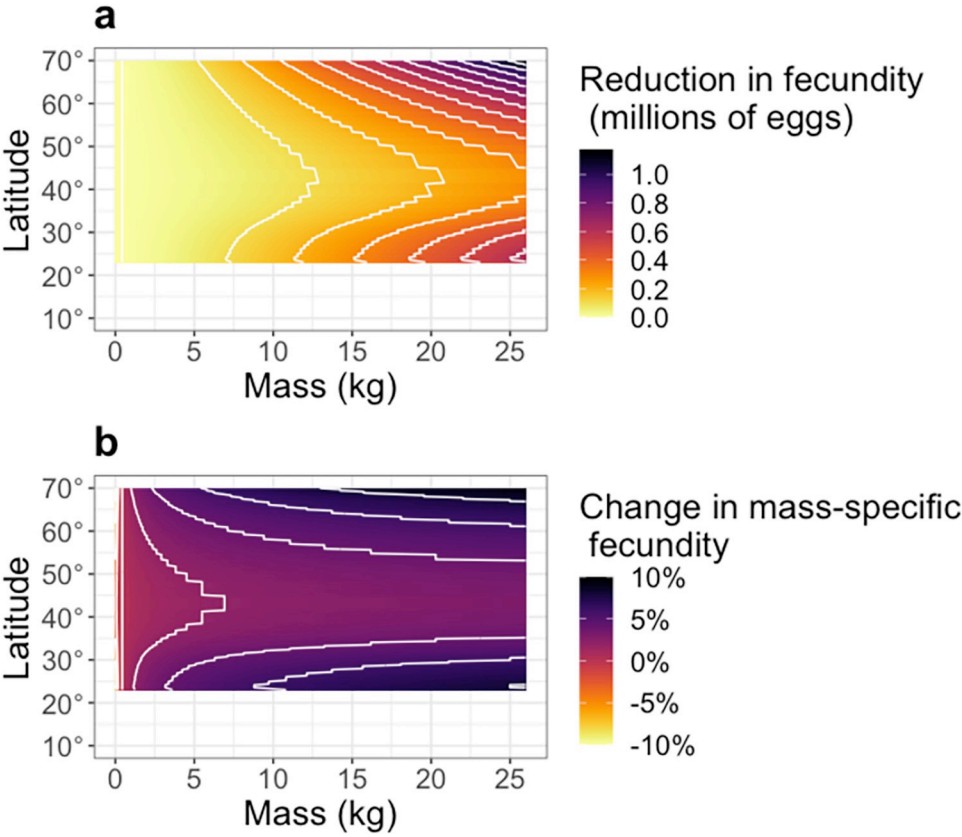

**Fig 4. Expected changes in mean fecundity with climate change (for a predicted 2.58˚C increase in SST at the end of the century under a high emissions scenario; [49]).** (a) Expected changes in the mean number of eggs across latitudes for different female sizes. The colours show the reduction in the expected number of eggs (in millions), with the smallest reduction in yellow and the largest in black. The white contour lines show reductions of 200,000 eggs. (b) Expected change in the mass-specific fecundity (in percentage) across latitudes for different female sizes. The colours show the reduction in the percentage change in mass-specific fecundity, with the largest increase in yellow and the largest decrease in black. The white contour lines show 2% changes in mass-specific fecundity. Current sea surface water temperature (SST) was assumed to be the historical mean (1981–2010) from the COBE-SST2 data provided by the NOAA/OAR/ESRL PSL (Boulder, Colorado, USA) (https://psl.noaa.gov/).

parameters: α (age at maturity) and $h$ (the rate at which the proportion of surplus energy invested into reproduction increases). Weight $w(t)$ is a function of time ($t$) and initially only depends on a growth parameter $k$ (not equivalent to the growth coefficient $K$ in the von Bertalanffy growth equation) and an offspring size parameter ($w_0$; theoretical weight at time = 0 as a proxy). We chose this model because it depends only on a few parameters for which we can get relatively reliable data from many fish species, yet it is able to accurately predict fish life histories. While more complex models might predict reproductive patterns for species or locations more accurately, increasing model complexity would have limited the number of species and locations for which we could find reliable estimates of all model parameters and thus would have come at the cost of generality.

Before maturity, the growth trajectory is described by the differential equation $dw/dt = P(w)$, where the production function $P(w)$ is assumed to be

$$P(w) = kw^{3/4} \tag{1}$$

After maturity, if $u(t)$ denotes the proportion of resources invested in growth $f(t)$, a proxy for fecundity, is given by

$$f(t) = (1 - u(t))P(w). \tag{2}$$

If we further assume that the function $u(t)$ is given by

$$u(t) = \begin{cases} 1 & ; t < \alpha \\ e^{-h(t-\alpha)} & ; t \geq \alpha \end{cases}, \tag{3}$$

we can solve for weight as a function of time $t$, knowing age at maturity $\alpha$, initial size $w_0$, and the rate at which resources diverted to reproduction change over time $h$:

$$w(t) = \begin{cases} \left(\dfrac{1}{4}kt + w_0^{1/4}\right)^4 & ; t < \alpha \\[3ex] \left[\dfrac{1}{4}k\left(\dfrac{1 - e^{-h(t-\alpha)}}{h} + \alpha\right) + w_0^{1/4}\right]^4 & ; t \geq \alpha \end{cases} \tag{4}$$

These equations assume that production scales at 3/4 with body size, but results are qualitatively consistent when a scaling equals to 2/3 (i.e., as assumed in Day and Taylor [9]; it is enough to change 4 with 3, and 1/4 with 1/3). In fact, Eq 4 can be adjusted to any scaling parameter of production.

To find the optimal energy allocation strategy for fish across latitudes, we predicted how demographic parameters change across latitudes using data from the literature. From the data (see data compilation and data analysis below), we obtained the relationships for natural mortality ($M$), $k$, and $w_0$ with latitude. Using these relationships, we looked at each latitude for the values of $\alpha$ that would maximise lifetime reproductive output ($R_0$) under a median value of $h$:

$$R_0 = (1 - p)e^{-M\alpha}\int_{\alpha}^{T} e^{-M(t-\alpha)}f(t)dt, \tag{5}$$

where $p$ is the initial burst of mortality (here arbitrarily fixed at 0.8; this parameter does not change optimization results), and $T$ is the longevity and was set to the age where only 1% of the individuals are expected to survive ($T = -\frac{1}{M}\ln\left(\frac{0.01}{1-p}\right)$).

We generated mass and fecundity relationships with the optimal age at maturity and optimal surplus energy allocation for each latitude. The reproductive scaling was then determined as the change in fecundity (log-scale) over the change in mass (log-scale) from maturity to $T$. Upon first glance, it would appear that fecundity should scale with mass at ¾, like production. However, changes in energy allocation from growth to fecundity also affect the scaling by diminishing the change in growth, and increasing the change in fecundity, allowing for hyperallometry.

To test the robustness of the model predictions, we implemented a second life history model proposed by Kozłowski [10] (see below). In this model, resource allocation also changes throughout ontogeny. The main difference with the Day and Taylor model is that Kozłowski's model assumes a seasonal environment and that changes in resource allocation are abrupt instead of gradual within each season starting from maturation; such bang-bang switching is optimal as proved by Kozłowski and Teriokhin [51]. Thus, each season is optimally divided between growth and reproduction. Numerical studies showed that results from Kozłowski's model can be well approximated by Day and Taylor equation (see S1 Supporting Information), and, therefore, simpler Day and Taylor's model was applied in the main body of the paper.

## Data compilation

To test whether the optimal reproductive schedules change across latitudes, we compiled data on location-specific mortality, growth, and initial size to calibrate the life history model (S7 Fig). For mortality, we compiled data on the rate of natural mortality ($M$) from literature, including datasets in [52–54]. We used only estimates of $M$ estimated from age-based catch curves or tag and release data (i.e., direct estimates) where the reference specified that the population was unfished or that fishing was insignificant, giving us a total of 131 observations for 74 species from 59 references. However, some references had multiple estimates of $M$ for the same species across a small range of latitudes. For those references, we grouped estimates of the same species and used the mean $M$ and the mean latitude instead, giving a total of 82 observations across latitudes 0˚ to 72.5˚.

To estimate the growth parameter $k$ and initial weight $w_0$, we compiled length at age data before maturity for 44 species from 32 references ($n = 49$), with latitudes ranging from 7.5˚ to 63˚. Length was converted into mass using length–weight relationships in FishBase that were either species specific or based on closely related species with similar body shape [55,56]. For each species at each location, we fitted Eq 4 (with $t<0$) to the data. Similarly, to restrict the range of the possible values of $h$ when optimising $R_0$ using Day and Taylor's model, we used the subset of these references where species approached their asymptotic mass to estimate $h$. In these instances, all mass at age data was used, and both Eq 2 and Eq 3 were fitted to the data. The subset of references for which $h$ was estimated included 28 species across latitudes ranging from 15.4˚ to 61˚. When optimizing maturity schedules in Day and Taylor's model, $h$ was fixed at its median value ($h = 0.26$; S8 Fig), but $\alpha$ was allowed to vary. Results are robust to fixing $h$ at its median observed value because variation in $R_0$ is consistently higher across values of $\alpha$ than across values of $h$ (S9 and S10 Figs).

Data on age at 50% maturity ($A_{50}$) were also compiled from published literature. Age at 50% maturity is the age at which half of the individuals are expected to be mature. Although often estimates of $A_{50}$ could be obtained indirectly when both length at 50% maturity ($L_{50}$) and the von Bertalanffy growth equation were known, we only used estimates of $A_{50}$ that were directly reported in the literature. The $R^2$ decreased by 42% if indirect estimates of $A_{50}$ were included ($R^2$: 0.33, CI: 0.21 to 0.43 if indirect estimates of $A_{50}$ were excluded versus $R^2$: 0.14, CI: 0.07 to 0.20 if indirect estimates of $A_{50}$ were included). We compiled a total of 127 observations of $A_{50}$ from 34 references and 59 species, with latitudes ranging from 9˚ to 73˚.

We used data on batch fecundity–female size relationships for marine fish from the published literature. Most of the data had been previously compiled by Barneche and colleagues [1], but we searched for additional studies, particularly at high latitudes and in underrepresented areas to have a broader geographical scope. Only marine species were considered for the analysis; species that migrate to or from freshwater were not included. Sharks and rays were also excluded since their fecundity was much lower than for the rest of the species because of their viviparity or large egg envelops, and tuna were excluded because their sizes were much larger than for the rest of the species. We used the number of eggs per reproductive pulse (i.e., batch fecundity) as the measurement for fecundity. While the literature also includes other measurements of fecundity, such as annual fecundity, such data are much less common and therefore were excluded from the analysis. Importantly, when the model included changes in seasonality across latitudes (i.e., longer periods favourable for reproduction at low versus high latitudes), age at maturity and reproductive scaling were still predicted to increase with latitude (S13 Fig). If a species had fecundity–female size data at one location for more than 1 year, only the year with the highest number of data points was included. To accurately estimate mass–fecundity relationships for each species, we excluded species for

which less than 10 data points per location were available. We compiled a total of 13,739 observations from 157 references and 153 species, with latitudes ranging from 0.9˚ to 74.5˚.

## Data analysis

All of the analyses were performed in R version 3.5.2 [57]. We used linear mixed effects models to estimate $M$, $k$, $w_0$, fecundity (i.e., number of eggs per batch), and $A_{50}$ using the package "brms" version 2.7.0 [58] assuming a Gaussian distribution.

We fitted a linear model to predict $M$ (in log-scale) as a function of absolute latitude (fixed effect) using only direct measurements of $M$. To predict $k$ and $w_0$ (both in log-scale), we fitted linear models with absolute latitude as a fixed effect. To estimate $k$ and $w_0$, we corrected for phylogenetic relatedness between species by including a random intercept of phylogeny with a covariance matrix reflecting the phylogenetic tree using the R package "rotl" [59]. For $M$, we chose not to correct for phylogenetic relatedness because our goal here was to understand the spatial distribution of different mortality rates or, in other words, what is the likely mortality rate of the "average fish" for a given latitude. We use predictions from these 3 models (S1 Table) to calibrate the life history model across latitudes. Life history models calibrated using the phylogenetically controlled estimates of $M$ yielded qualitatively similar predictions (i.e., an increase in age at maturity and reproductive scaling with increasing latitude).

The model estimating fecundity (number of eggs or offspring per batch, in log-scale) included female weight (g, in log-scale), latitude (absolute value), and their interactions as fixed effects, assuming a Gaussian distribution. Here, the reproductive scaling is the slope of the regression between natural logarithm of fecundity and the natural logarithm of female weight. Since each species had multiple observations at each location, species was included as a random effect that could modify the intercept and the slope of relationship between fecundity and female weight. In addition, a random intercept was included for phylogeny, associated to a variance–covariance that corrected for phylogenetic relatedness.

The model estimating $A_{50}$ (years, in log-scale) included latitude as a fixed effect and the random intercept correcting for phylogenetic relatedness. The model was fitted assuming a Gaussian distribution.

## Predicted size-specific changes in reproductive output with climate change

To estimate the expected change in fecundity for a given female mass with climate change, we used predicted increases in water temperature under different climate change scenarios: a low emissions scenario (RCP2.6) and a high emissions scenario (RCP8.5) [49]. We estimated mean sea surface water temperature (SST) (1981 to 2010) across the globe from the *COBE-SST2* data provided by the NOAA/OAR/ESRL PSL (Boulder, Colorado, USA), from their website at https://psl.noaa.gov/. Using these data, we fitted a general additive model (GAM) to predict SST as a function of absolute latitude (S14 Fig).

At each latitude, we calculated the predicted SST mid-century (2031 to 2050) and late-century (2081 to 2100) for the low and high emissions scenarios predicted by the IPCC (an increase of 0.64˚C and 0.73˚C for the mid- and late-century for the low emissions scenario, and a 0.95˚C and a 2.58˚C for the high emissions scenario). The IPCC predictions were relative to temperature in 1986 to 2005, which is within the time interval when most of the fish reproductive data were collected (S15 Fig). For each scenario at each latitude, we calculated the predicted mid- and late-century SST, and we found the latitude that currently experienced that mean temperature. Then, assuming fish reproductive scaling responds to changes in temperature, we estimated future reproduction by using the fitted mass–fecundity relationship for the latitude that matched the predicted SST. At low latitudes, an increase in SST was unmatched

(e.g., for any future temperature >27°C, there were no latitudes that were currently experiencing that mean temperature), and, therefore, changes in reproduction could not be predicted.

## Supporting information

**S1 Fig. Demographic parameters across latitudes.** The grey points show data points. The solid lines show models' predictions, and the dashed lines show the 95% credible intervals. (**a**) $M$ as a function of latitude ($\mathrm{Ln}(M) = a + b *$ latitude (absolute value), where $a = -0.31$ (95% CI: −0.82 to 0.20) and $b = -0.02$ (95% CI: −0.04 to −0.01); S1 Table). The underlying data for this panel can be found in S3 Data. (**b**) $k$ as a function of latitude ($\mathrm{Ln}(k) = a + b *$ latitude (absolute value), where $a = 1.47$ (95% CI: 0.66 to 2.30) and $b = -0.01$ (95% CI: −0.03 to 0.01); S1 Table). (**c**) $w_0$ as a function of latitude ($\mathrm{Ln}(w_0) = a + b *$ latitude (absolute value), where $a = 2.67$ (95% CI: 0.63 to 5.05) and $b = 0.00$ (95% CI: −0.05 to 0.04); S1 Table). The underlying data for panels (**b**) and (**c**) can be found in S4 Data.
(TIFF)

**S2 Fig. Effects of size-dependent mortality on model predictions.** (**a**) Relative mortality rate $M$ as a function of relative mass. (**b**) Optimal age at maturity ($\alpha$) across latitudes. (**c**) Reproductive scaling across latitudes. The colours show the level of size dependency in mortality ($q$), going from size independent (purple) to strongly size dependent (yellow). Predictions underlying this figure assume changes in fecundity ($F$) across latitudes following the Equation in Fig 2 and S1 Table.
(TIFF)

**S3 Fig. Intraspecific patterns in reproductive scaling.** Reproductive scaling is shown for the highest and lowest latitude populations for species with reproductive data spanning >20° latitude. The error bars show the 95% credible intervals, and the colours indicate absolute latitude. The dashed black line shows reproductive isometry. The reproductive scaling for *Engraulis ringens* is 0.86 (95% CI: 0.50 to 1.20) and 1.10 (95% CI: 0.68 to 1.44) at absolute latitudes of 9° and 40°, respectively. The reproductive scaling for *Gadus morhua* is 1.19 (95% CI: 0.87 to 1.59) and 1.41 (95% CI: 1.05 to 1.90) at absolute latitudes of 43.5° and 68°, respectively. The reproductive scaling for *Katsuwonus pelamis* is 0.83 (95% CI: 0.47 to 0.18) and 1.07 (95% CI: 0.74 to 1.42) at absolute latitudes of 5° and 36°, respectively. The reproductive scaling for *Reinhardtius hippoglossoides* is 1.17 (95% CI: 0.83 to 1.54) and 1.37 (95% CI: 0.95 to 1.84) at absolute latitudes of 48° and 74.5°, respectively. The reproductive scaling for *Scomberomorus cavalla* is 0.82 (95% CI: 0.46 to 1.17) and 1.07 (95% CI: 0.73 to 1.44) at absolute latitudes of 4° and 35°, respectively. The reproductive scaling for *Sebastes mentella* is 1.21 (95% CI: 0.83 to 1.58) and 1.40 (95% CI: 0.92 to 1.84) at absolute latitudes of 48.5° and 70°, respectively.
(TIFF)

**S4 Fig. Simulated reproductive scaling across latitudes assuming different overhead costs to reproduction (cr).** Panels (**a**-**c**) show 3 different relationships between the overhead cost of reproduction and female size. Panels (**d**-**f**) show the reproductive scaling predicted by the life history optimisation model given different reproductive costs. Panels (**a**) and (**d**) assume no overhead cost to reproduction ($cr = 0$). Panels (**b**) and (**e**) assume an overhead cost of reproduction that increases slightly with female size ($cr = 0.4/(1 + e^{-0.002*(w-2000)})$, where $w$ is female weight). Panels (**c**) and (**f**) assume a steep increase in overhead cost of reproduction with female size ($cr = 0.8/(1 + e^{-0.001*(w-1000)}) - 0.15$). Predictions for reproductive scaling were generated assuming fecundity was given by: $f(t) = (1 - u(t))P(w)*(1 - cf)$, where $f(t)$ was optimised given the latitudinal gradient for $M$, $k$, and $w0$ (S1 Table). Because panel (**a**) represents the

scenario where reproduction has no overhead costs, S4A Fig is identical to Fig 2A.
(TIFF)

**S5 Fig. Expected changes of the mean number of eggs (in millions) produced for a female of a given mass at a given latitude under predicted climate change scenarios.** RCP2.6 and RCP8.5 scenarios are a low- and very high-emission scenario, respectively. Expected fecundity changes across latitudes for different female sizes for 2031–2050 under the RCP2.6 scenario, 0.64˚C increase in SST (**a**), for 2081–2100 under the RCP2.6 scenario, 0.73˚C increase in SST (**b**), for 2031–2050 under the RCP8.5 scenario, 0.95˚C increase in SST (**c**), and for 2081–2100 under the RCP8.5 scenario, 2.58˚C increase in SST (**d**). The colours show the reduction in the expected number of eggs, with the smallest reduction in yellow and the largest in black. The white contour lines show consecutive reductions in fecundity of 100,000 eggs. Current sea surface water temperature (SST) was assumed to be the historical mean (1981–2010) from the COBE-SST2 data provided by the NOAA/OAR/ESRL PSL (Boulder, Colorado, USA) (https://psl.noaa.gov/).
(TIFF)

**S6 Fig. Expected change in the mass-specific fecundity (in percentage) for a female of a given mass at a given latitude under predicted climate change scenarios.** The colours show the reduction in the percentage change in mass-specific fecundity, with the largest increase in yellow and the largest decrease in black. RCP2.6 and RCP8.5 scenarios are a low- and very high-emission scenario, respectively. (**a**) Expected change in the mass-specific fecundity across latitudes for different female sizes for 2031–2050 under the RCP2.6 scenario (0.64˚C increase in SST). (**b**) Expected change in the mass-specific fecundity across latitudes for different female sizes for 2081–2100 under the RCP2.6 scenario (0.73˚C increase in SST). (**c**) Expected change in the mass-specific fecundity across latitudes for different female sizes for 2031–2050 under the RCP8.5 scenario (0.95˚C increase in SST). (**d**) Expected change in the mass-specific fecundity across latitudes for different female sizes for 2081–2100 under the RCP8.5 scenario (2.58˚C increase in SST). The colours show the change in the mass-specific fecundity, with the largest increase in percentage of mass-specific fecundity in yellow and the largest decrease in black. The white contour lines show consecutive 2% changes in mass-specific fecundity. Current sea surface water temperature (SST) was assumed to be the historical mean (1981–2010) from the COBE-SST2 data provided by the NOAA/OAR/ESRL PSL (Boulder, Colorado, USA) (https://psl.noaa.gov/).
(TIFF)

**S7 Fig. Schematic representation of our approach.** First, we compile life history data (mortality, growth, and offspring size) of marine fishes across latitudes from the literature. We used these compiled data to fit statistical models and predict life history as a function of latitude. Then, using those predictions, we found the age at maturity that would maximise lifetime reproductive output ($R_0$) at each latitude, as well as the resulting reproductive scaling. Finally, we compared those predictions to data from the literature.
(TIFF)

**S8 Fig. Distribution of estimated $h$ values obtained by fitting Day and Taylor's (1997) model to mass and age data of 29 fish populations.** The median value equals 0.26 (first quartile is 0.18, third quartile is 0.39, mean is 0.48 ± 0.14 SE).
(TIFF)

**S9 Fig. Predicted $R_0$ across a range of ages at maturity ($\alpha$) and $h$ values at 0˚, 30˚, and 60˚ latitude (panels a, b, and c, respectively).** The colours show the predicted value of $R_0$ from

low (blue) to high (yellow). Panel (**d**) shows the distribution of standard deviations in $R_0$ for constant $h$ values but varying $\alpha$ values (labelled as $\alpha$, in orange), and for constant $\alpha$ values but varying $h$ values (labelled as $h$, in grey) at 0˚, 30˚, and 60˚ latitude. The standard deviation of $R_0$ with variations in $\alpha$ ranges from 1.13 to 8.78 (median of 8.32) at 0˚ latitude, from 9.67 to 28.9 (median of 24.1) at 30˚ latitude, and from 28.21 to 124.52 (median 121.69) at 60˚ latitude. The standard deviation of $R_0$ with variations in $h$ ranges from 0.43 to 3.55 (median of 2.51) at 0˚ latitude, from 10.88 to 18.33 (median of 13.91) at 30˚ latitude, and from 29.28 to 53.80 (median 35.14) at 60˚ latitude.
(TIFF)

**S10 Fig.** Life history predictions of the optimal reproductive schedules for the lower ($h = 0.18$ in panels (**a**) and (**b**)) and the upper quartile of the distribution of $h$ values estimated from the mass at age relationship data ($h = 0.39$ in panels (**c**) and (**d**)). (**a** and **c**) Optimal $\alpha$ across latitudes. (**b** and **d**) Reproductive scaling resulting from the optimal reproductive schedule. The black lines show the medians of the posterior distributions, and the grey ribbons show the 95% credible intervals. The life history optimisation assumes mortality ($M$), growth ($k$), and offspring size ($w_0$) change across latitudes following: $Ln(D) = a+b*latitude$, where D is the demographic rate. For mortality, $a = -0.31$ (95% CI: $-0.82$ to 0.20) and $b = -0.02$ (95% CI: $-0.04$ to $-0.01$). For growth, $a = 1.47$ (95% CI: 0.66 to 2.30) and $b = -0.01$ (95% CI: $-0.03$ to 0.01)). For offspring size, $a = 2.67$ (95% CI: 0.63 to 5.05) and $b = 0.00$ (95% CI: $-0.05$ to 0.04). All demographic estimates can be found in S1 Table.
(TIFF)

**S11 Fig. Optimal growth trajectory resulting from Kozłowski's model (dots) and fitted Day and Taylor's equation (Eq 3; solid line) with the same initial size $w_0$ (= 1), growth rate parameter $k$ (= 5) and age at maturity.** Yearly mortality rate equals 0.2 (**a**), 0.6 (**b**), and 1.0 (**c**). Optimal age at maturity equals 11 years (**a**), 2.88 (**b**), and 1.00 years (**c**). Productive season of the length 200 days was assumed in Kozłowski's model.
(EPS)

**S12 Fig. Length of the reproductive season (in days) across latitudes.** The grey points show data points corrected for phylogenetic effects. The solid lines show the model's predictions, and the dashed lines show the 95% credible intervals. The data underlying this figure can be found in S5 Data. The model's predictions are: $P = a+b*latitude$, where $P$ is the season length (as proportion of the year), $a = 2.96$ (95% CI: 2.35 to 3.59) and $b = -0.07$ (95% CI: $-0.08$ to $-0.05$), with a logit link.
(TIFF)

**S13 Fig. Optimal reproductive schedules predicted by Kozłowski's life history model assuming b = 3/4.** The grey points show the optimal value at each latitude. The black lines show the linear fits. (**a**) Optimal age at maturity (years) across latitudes. Discontinuities appear when age at maturity jumps to a next year. (**b**) Reproductive scaling resulting from the optimal reproductive schedule. The life history optimisation assumes mortality ($M$), changes across latitudes following: $Ln(M) = a+b*latitude$, where $a = -0.31$ (95% CI: $-0.82$ to 0.20) and $b = -0.02$ (95% CI: $-0.04$ to $-0.01$) (S1 Table). The model also assumes season length ($P$, as proportion of the year), changes across latitudes following: $Ln(P) = a+b*latitude$, where $a = 2.96$ (95% CI: 2.35 to 3.59) and $b = -0.07$ (95% CI: $-0.08$ to $-0.05$), with a logit link.
(TIFF)

**S14 Fig. Mean sea surface temperature (SST) for 1981–2010 across the globe (in a 1˚ × 1˚ grid) as a function of absolute latitude.** The black points show the data points, and the red

line shows the model fit (from a general additive model, GAM). SST values were extracted from the COBE-SST2 data provided by the NOAA/OAR/ESRL PSL (Boulder, Colorado, USA) (https://psl.noaa.gov/).
(TIFF)

**S15 Fig. Distribution of years when the fish fecundity–mass data were collected.** The box-plot shows the upper (2000) and lower (1984) 25% quantiles, and the horizontal black line shows the median (1991).
(TIFF)

**S1 Table. Fitted model predictions and 95% credible intervals for $M$, $k$, and $w_0$ as a function of latitude.**
(DOCX)

**S1 Data. Compiled data on age at 50% maturity for marine fishes.** The dataset contains information on species, length at 50% maturity (in cm, "$L50\_cm$") and its corresponding length type (TL = total length, SL = standard length, FL = fork length), total length at 50% maturity (in cm, "$L50\_TL$"), mass at 50% maturity (in g, "$M50\_g$"), age at 50% maturity (years, "$Age\_50$") and whether it was directly reported on the reference ("$Age\_50\_Reference$"), type of estimate, reference, location, latitude, longitude, and date.
(CSV)

**S2 Data. Compiled data on size–fecundity relationships for marine fishes.** The dataset contains information on order, family, species, female mass (in grams, "$Mass\_g$"), fecundity ("$Fecundity.\_n.Of.Eggs$") and their units, the natural logarithm of female mass ("$lnMass$"), the natural logarithm of fecundity ("$lnFecundity$"), location of data collection, latitude (and its absolute value), longitude, date, reference, and data source (i.e., in which study the data was compiled).
(CSV)

**S3 Data. Compiled data on direct estimates of natural mortality of marine fishes.** The dataset contains information on species, estimates of natural mortality ("$M$"), the natural logarithm of natural mortality ("$LnM$"), absolute latitude, number of subpopulations that were grouped together ("$n$"), the range of latitudes and longitudes that these subpopulations encompassed, and the reference.
(CSV)

**S4 Data. Estimates on the growth ($k$) and offspring size ($w_0$) parameters from Day and Taylor's (1997) model (assuming a production scaling of ¾) from size at age data on marine fishes.** The dataset contains information on species, reference, location, latitude (and absolute latitude), longitude, date, $k$, $w_0$, and the natural logarithms of $k$ and $w_0$ ("$LnK$", "$LnW0$").
(CSV)

**S5 Data. Compiled data on the length of the reproductive season for populations of marine fishes.** The dataset includes information on species, season length and its corresponding units, season type, location, latitude, longitude, and reference.
(CSV)

**S1 Supporting Information. Supporting information that includes the following: (1) a sensitivity analysis of size-dependent mortality; (2) an exploration of the effects of overhead costs of reproduction in reproductive scaling; (3) a sensitivity analysis of size-specific changes in reproductive output with climate change; and (4) an analysis of optimal energy**

**allocation following Kozłowski's life history model.**
(DOCX)

**S1 Compiled Data References. References for the compiled data found in S1–S5 Data files.**
(PDF)

## Acknowledgments

We thank Michaela Parascandalo for compiling the fecundity data.

## Author Contributions

**Conceptualization:** Mariana Álvarez-Noriega, Craig R. White, Jan Kozłowski, Dustin J. Marshall.

**Data curation:** Mariana Álvarez-Noriega, Dustin J. Marshall.

**Formal analysis:** Mariana Álvarez-Noriega, Jan Kozłowski.

**Funding acquisition:** Craig R. White, Dustin J. Marshall.

**Methodology:** Mariana Álvarez-Noriega, Jan Kozłowski, Troy Day, Dustin J. Marshall.

**Supervision:** Dustin J. Marshall.

**Validation:** Jan Kozłowski, Troy Day.

**Visualization:** Mariana Álvarez-Noriega, Jan Kozłowski.

**Writing – original draft:** Mariana Álvarez-Noriega.

**Writing – review & editing:** Craig R. White, Jan Kozłowski, Troy Day, Dustin J. Marshall.

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
