## [Editor Report · Decision Letter 0]

19 Dec 2022

Dear Dr Alvarez-Noriega, 

Thank you for submitting the new version of your manuscript entitled "Reproductive scaling and maturation are the product of life history optimisation in fishes" for consideration as a Research Article by PLOS Biology.

Your revisions have now been evaluated by the PLOS Biology editorial staff, and I'm writing to let you know that we would like to send your submission out for re-review.

IMPORTANT: We note that you have not included a marked up version of your manuscript, indicating the changes made since the previous version; please could you upload this as an additional file when you upload the additional metadata? (see next paragraph)

Once your full submission is complete, your paper will undergo a series of checks in preparation for re-review. After your manuscript has passed the checks it will be sent out for review. To provide the metadata for your submission, please Login to Editorial Manager (https://www.editorialmanager.com/pbiology) within two working days, i.e. by Dec 21 2022 11:59PM.

Kind regards,

Roli Roberts

Roland Roberts, PhD

Senior Editor

PLOS Biology

rroberts@plos.org

---

## [Decision Letter · Decision Letter 1]

6 Feb 2023

Dear Dr Alvarez-Noriega,

Thank you for your patience while we considered your revised manuscript "Reproductive scaling and maturation are the product of life history optimisation in fishes" for consideration as a Research Article at PLOS Biology. Your revised study has now been evaluated by the PLOS Biology editors, the Academic Editor and two of the original reviewers.

IMPORTANT: You'll see that while reviewer #1 is now mostly satisfied, reviewer #3 retains two concerns that will have to be addressed before further consideration. The Academic Editor agrees with these points, and has kindly provided some additional guidance (see the foot of this letter) which you may find helpful.

In light of the reviews, which you will find at the end of this email, we are pleased to offer you the opportunity to address the remaining points from the reviewers in a revision that we anticipate should not take you very long. We will then assess your revised manuscript and your response to the reviewers' comments with our Academic Editor aiming to avoid further rounds of peer-review, although might need to consult with the reviewers, depending on the nature of the revisions.

**IMPORTANT - SUBMITTING YOUR REVISION**

*Resubmission Checklist*

*Published Peer Review*

*PLOS Data Policy*

*Blot and Gel Data Policy*

Sincerely,

Roli Roberts

Roland Roberts, PhD

Senior Editor

PLOS Biology

rroberts@plos.org

REVIEWS:

Reviewer #1:

The authors have addressed my questions (major and minor) pretty well. I am glad to see the newly added Fig. 3 and the text around it, which help a lot. 

I only have a very minor concern here, and the authors can ignore it, if they don't agree with me. In lines 156-158, the authors said: "Our findings, in conjunction with others (11), imply that many of the life history patterns we observe in nature may be driven by the evolutionary optimisation of life histories, rather than mechanistic constraints." 

I understand what they mean, and I agree with them on what they said after this sentence: "life history optimisation approaches may be a useful tool for..."

But, the first sentence sounds like evolutionary optimization and mechanistic constraints clash, but they often work in concerted ways. As the same authors said in a cited paper (Ref.11, by the way, it is a brilliant paper): "Our approach does not assume that life is unconstrained by physics and chemistry, but instead assumes that there is substantial (and underappreciated) opportunity for evolutionary optimization within these constraints."

Anyway, this is very minor, just about wording. 

Congratulations on a great paper.

Reviewer #3:

General comments:

This revised manuscript deals with an important topic: how do reproductive scaling relationships change with latitude. Although we already know much about how life-history traits vary with latitude in a variety of organisms, we know relatively little about how their body-mass scaling varies with latitude. The authors have extensively revised their manuscript in many useful ways to make it stronger, but I still have two major concerns that I believe the authors should address.

(1) The authors have not acknowledged the model of Jeffery Arendt, which makes similar predictions to those presented in this manuscript (see e.g., Arendt 2011, 2015; Arendt & Fairbairn 2012). Arendt's model is based on reported observations that size-fecundity relationships are steeper in colder than warmer environments, a pattern that is consistent with the latitudinal trend predicted and observed by the authors. Based on these temperature-dependent size-fecundity relationships, Arendt predicts that ectotherm animals should mature later at larger sizes in colder environments (i.e., the temperature-size rule), as does the authors' model. The predictions of Arendt's model appear to be similar to those of the authors' model, at least in part, and should be cited. It would be useful to compare the details of these two models. How are they similar/different? It would also be worthwhile to point out that Arendt has found both positive (Arendt 2015) and negative (Arendt & Fairbairn 2012) evidence supporting his model. 

(2) I am worried that the observed data show that the log-linear mass-fecundity scaling slope increases only slightly with increasing latitude (by only ~0.25), whereas the scaling slope predicted by their model changes a huge amount (by ~4.5: Figure 2) that is almost 20 times greater. The authors point out this discrepancy between model predictions and observed data, but I feel that they should make clear that this discrepancy is huge, thus weakening one's confidence in the model. Note also that the model predictions show that the log-linear scaling slope should be as high as 7 at high latitudes, which would indicate that fecundity increases by 7 orders of magnitude (10,000,000 times) with only a 1 order of magnitude (10 times) increase in body mass. This seems unrealistic. Therefore, something appears to be fundamentally wrong with (or missing from) the authors' model. Although the authors state that it is remarkable that their model correctly predicts an increase in the size-fecundity scaling slope with increasing latitude, this does not change the fact that it makes highly erroneous and unrealistic quantitative predictions of this change. In a review of mechanisms affecting the temperature-size rule, Verberk et al. (2021) suggest that other environmental factors, such as resource supply, may significantly relationships between size, fecundity and temperature. Perhaps the authors' model would be improved by focusing on the effects of specific environmental factors, such as temperature, resource supply, etc., rather than 'latitude' which represents a collage of variable environmental factors. 

Specific comments:

L 201-202: In my opinion, steeper size-fecundity scaling with increasing latitude should not be considered "unexpected" because size-fecundity scaling exponents have been reported to increase with decreasing temperature in some animal species (e.g., McCabe & Partridge 1997; Scheiner & Yamplosky 1998; Weetman & Atkinson 2004; Arendt 2015; but see Arendt & Fairbairn 2012), and temperature correlates with latitude.

L 210-214: This is not an entirely novel prediction: see Arendt (2011). 

Literature cited:

Arendt, J. D. (2011). Size‐fecundity relationships, growth trajectories, and the temperature‐size rule for ectotherms. Evolution: International Journal of Organic Evolution, 65(1), 43-51.

Arendt, J. (2015). Why get big in the cold? Size-fecundity relationships explain the temperature‐size rule in a pulmonate snail (Physa). Journal of Evolutionary Biology, 28(1), 169-178.

Arendt, J., & Fairbairn, D. (2012). Reproductive allometry does not explain the temperature-size rule in water striders (Aquarius remigis). Evolutionary Ecology, 26(3), 745-757.

McCabe, J., and L. Partridge. 1997. An interaction between environmental temperature and genetic variation for body size for the fitness of adult female Drosophila melanogaster. Evolution 51:1164-1174.

Scheiner, S. M., and L. Y. Yampolsky. 1998. The evolution of Daphnia pulex in a temporally varying environment. Genet. Res. 72:25-37.

Verberk, W. C., Atkinson, D., Hoefnagel, K. N., Hirst, A. G., Horne, C. R., & Siepel, H. (2021). Shrinking body sizes in response to warming: explanations for the temperature-size rule with special emphasis on the role of oxygen. Biological Reviews, 96(1), 247-268.

Weetman, D., & Atkinson, D. (2004). Evaluation of alternative hypotheses to explain temperature-induced life history shifts in Daphnia. Journal of Plankton Research, 26(2), 107-116.

COMMENTS FROM THE ACADEMIC EDITOR (lightly edited):

I have now read in detail the authors' responses to comments from the initial version, the revised version, and both reviewers' reports. I fully share the general viewpoint that this revised version has improved a lot by nicely accounting for several problems identified in the previous version. While one reviewer is satisfied with the revised version (and only pointed out a potentially misleading sentence), the other reviewer found two major problems with this revised version.

First, previous work by Arendt, who proposed a model making a similar prediction as the authors do here (i.e. that size-fecundity relationships should be steeper in colder than warmer environments, has to be discussed as its existence might question the novelty of the authors' model.

Second, the huge overestimation of the scaling slope increase with latitude when using the author’s model (i.e. a twenty fold overestimation), which might question the consequences for quantitative predictions and practical applications in terms of management, have to be discussed as a potential limitation of this work. From my own detailed reading, I fully share these two major concerns.

Both issues have to be thoroughly discussed to better assess the novelty and usefulness of the model proposed by the authors. Moreover, while I appreciated the link the authors explicitly did between their findings and the existence of a slow-fast continuum of life histories, I found it too superficial and ambiguous. Thus, their findings that the pace of life of marine fish is faster at low latitudes than at high latitude matches previous findings about variation in the pace of life in relation to elevation, but is opposite to previous findings that birds have a slower pace of life in tropical areas than in temperate areas (see e.g. Wiersma et al. 2007 PNAS). A more thorough discussion of the findings reported here and the universal existence of a slow-fast continuum of life histories (at least across species within vertebrate classes) is required.

---

## [Editor Report · Decision Letter 2]

22 Mar 2023

Dear Dr Alvarez-Noriega,

Thank you for your patience while we considered your revised manuscript "Reproductive scaling and maturation are the product of life history optimisation in fishes" for publication as a Research Article at PLOS Biology. This revised version of your manuscript has been evaluated by the PLOS Biology editors and the Academic Editor.

Based on our Academic Editor's assessment of your revision, we are likely to accept this manuscript for publication, provided you satisfactorily address the remaining points raised by the Academic Editor. Please also make sure to address the following data and other policy-related requests:

IMPORTANT - Please attend to the following:

a) Please address the remaining requests from the Academic Editor (see the bottom of this letter).

b) We struggled a bit with the Title - normally I'd suggest something, but it's currently eluding me, so I'll pass it back to you! To me, the phrase "reproductive scaling" (dependence of fecundity on body mass?) is not very intuitive; also the study is largely about latitude and the potential consequences of climate change, neither of which are reflected here. To my mind, what the title actually means is "the dependence of fish reproduction on size ('reproductive scaling') and age ('maturation') is driven by a number of factors, but mainly latitude, with problematic consequences in the face of climate change" - could you possibly fashion a more accessible/appealing title, which encapsulates what you did, what you found, and its implications? We like titles to be declarative, with an active verb and no punctuation, if possible. I'm happy to discuss this further...!

c) Please provide a blurb, according to the instructions in the submission form.

d) Please address my Data Policy requests below; specifically, we need you to supply the numerical values underlying Figs 1AB, 2ABC, 3C, 4AB, S1ABC, S2ABC, S3, S4ABCDEF, S5ABCD, S6ABCD, S8, S9ABCD, S10ABCD, S11ABC, S12, S13AB, S14, S15, either as a supplementary data file or as a permanent DOI’d deposition. I note that you already have a zipped supplementary data folder, but the data in these files looks rather “raw” to me, and we do need the values displayed in the Figs and/or any code needed to generate them.

e) Please cite the location of the data clearly in all relevant main and supplementary Figure legends, e.g. “The data underlying this Figure can be found in S1 Data” or “The data underlying this Figure can be found in https://doi.org/10.5281/XXXXX"

We expect to receive your revised manuscript within two weeks. 

*Published Peer Review History*

*Press*

Sincerely,

Roli

Roland Roberts, PhD

Senior Editor,

rroberts@plos.org,

PLOS Biology

DATA POLICY:

Regardless of the method selected, please ensure that you provide the individual numerical values that underlie the summary data displayed in the following figure panels as they are essential for readers to assess your analysis and to reproduce it: Figs 1AB, 2ABC, 3C, 4AB, S1ABC, S2ABC, S3, S4ABCDEF, S5ABCD, S6ABCD, S8, S9ABCD, S10ABCD, S11ABC, S12, S13AB, S14, S15. NOTE: the numerical data provided should include all replicates AND the way in which the plotted mean and errors were derived (it should not present only the mean/average values).

DATA NOT SHOWN?

COMMENTS FROM THE ACADEMIC EDITOR (lightly edited):

This revised version nicely accounts for all remaining problems one reviewer and myself identified in the previous version. In particular, the revised MS provides both a thorough discussion of the model proposed by Arendt and a convincing explanation (involving the metric used to measure the reproductive output and confounding effect of reproductive costs) for the mismatch between the expected and observed scaling coefficients.

I also enjoyed the new and more thorough discussion about the slow-fast continuum of life histories. However, although it is still speculative, I found the authors' interpretation that mortality risks rather than latitude per se drive life history covariations highly interesting and worth explicitly presenting in this work (after lines 160-163).

In addition I found the following details to fix:

l. 132: "allocation", not "investment"

l. 136: "allocated", not "invested"

l. 161: "are driven" instead of "is driven"

l. 182: "allocation", not "investment"

---

## [Editor Report · Decision Letter 3]

6 Apr 2023

Dear Dr Alvarez-Noriega,

Thank you for the submission of your revised Research Article "Life-history optimisation drives latitudinal gradients and responses to global change in marine fishes" for publication in PLOS Biology. On behalf of my colleagues and the Academic Editor, Jean-Michel Gaillard, I'm pleased to say that we can in principle accept your manuscript for publication, provided you address any remaining formatting and reporting issues. These will be detailed in an email you should receive within 2-3 business days from our colleagues in the journal operations team; no action is required from you until then. Please note that we will not be able to formally accept your manuscript and schedule it for publication until you have completed any requested changes.

Sincerely, 

Roli Roberts

Senior Editor

PLOS Biology

rroberts@plos.org